# Traffic-Related High Sleep Disturbance in the LIFE-Adult Cohort Study: A Comparison to the WHO Exposure-Response-Curves

**DOI:** 10.3390/ijerph20064903

**Published:** 2023-03-10

**Authors:** Melanie Schubert, Karla Romero Starke, Julia Gerlach, Matthias Reusche, Pauline Kaboth, Wolfram Schmidt, Dieter Friedemann, Janice Hegewald, Hajo Zeeb, Andrea Zülke, Steffi G. Riedel-Heller, Andreas Seidler

**Affiliations:** 1Institute and Policlinic of Occupational and Social Medicine, Faculty of Medicine, Technische Universität Dresden, 01307 Dresden, Germany; 2Institute of Transport Planning and Road Traffic, Technische Universität Dresden, 01069 Dresden, Germany; 3Institute for Medical Informatics, Statistics and Epidemiology, University of Leipzig, 04107 Leipzig, Germany; 4Lohmeyer GmbH, 01067 Dresden, Germany; 5CDF, Schallschutz GmbH, 01108 Dresden, Germany; 6Department of Prevention and Evaluation, Leibniz-Institute for Prevention Research and Epidemiology—BIPS, 28359 Bremen, Germany; 7Health Sciences Bremen, University of Bremen, 28359 Bremen, Germany; 8Institute of Social Medicine, Occupational Health and Public Health, University of Leipzig, 04103 Leipzig, Germany

**Keywords:** self-reported high sleep disturbance, HSD, traffic noise, road, rail, air, WHO environmental noise guideline

## Abstract

Sleep is negatively affected by environmental noise. In the present study, we investigated self-reported high sleep disturbances (being “highly sleep disturbed”—HSD) from road traffic (primary and secondary road networks), rail (train and tram) and air traffic noise in the LIFE-Adult cohort study in Leipzig, Germany. For this, we used exposure data from 2012 and outcome data of Wave 2 (collected during 2018–2021). HSD was determined and defined according to internationally standardized norms. The highest risk for transportation noise-related HSD was found for aircraft noise: the odds ratio (OR) was 19.66, 95% CI 11.47–33.71 per 10 dB increase in L_night_. For road and rail traffic, similar risk estimates were observed (road: OR = 2.86, 95% CI 1.92–4.28; rail: OR = 2.67, 95% CI 2.03–3.50 per 10 dB L_night_ increase). Further, we compared our exposure-risk curves with the curves of the WHO environmental noise guidelines for the European region. The proportion of individuals with HSD for a given noise level was lower for rail traffic but higher for aircraft noise in the LIFE study than in the WHO curves. For road traffic, curves are not directly comparable because we also included the secondary road network. The results of our study add to the body of evidence for increased health risks by traffic noise. Moreover, the results indicate that aircraft noise is particularly harmful to health. We recommend reconsidering threshold values for nightly aircraft exposure.

## 1. Introduction

Sleep is one of the major determinants of health and wellbeing in humans. Thus, sleep disturbances might affect all major body systems, such as the immune, endocrine and the cardiovascular system, and increase the risk of several diseases [1,2,3,4].

Sleep is regulated by homeostatic and circadian processes [4] and is affected by lifestyle, environmental and psychological factors [3,5,6,7]. Lifestyle factors potentially detrimental to sleep may include alcohol drinking, smoking and electronic media use, while psychological factors may involve work stress, worry and rumination. Environmental factors comprise exposure to light, environmental noise and the work environment (in particular, nightshifts).

According to the European Environmental Agency (EEA) about 6.5 million people suffer from environmental noise-related chronic high sleep disturbance [8], which is considered one of the most important non-auditory effects of environmental noise [9]. The World Health Organization (WHO) estimated in 2011 that 903,000 disability-adjusted life years (DALYs) are lost annually from environmental noise-related sleep disturbances in Europe [9]. More recently, Eriksson and colleagues [10] estimated the proportion of “highly sleep disturbed” (HSD) and the burden of disease from road and rail traffic in Sweden. In line with the WHO, the authors found that traffic noise-induced sleep disturbances were the most important contributor to the burden of disease. Other current studies show high burden of disease due to HSD from road traffic noise for urban complexes in Thessaloniki-Napoli [11] and the Frankfurt Main area [12].

In 2019, the WHO published the “Environmental noise guidelines for the European region” [13]. These guidelines included a systematic review with a meta-analysis on the noise-related effects on sleep [14]. The literature search was conducted from 2000 to 2015. In their analysis, the authors differentiated between questions that asked about the particular effects of a specific noise source and those not referring to a noise source in their question. For questions referring to a specific noise source, 12 epidemiological studies were included for road traffic noise, 5 studies for rail traffic noise and 6 studies for aircraft noise. These studies were predominately from Asia (Vietnam, *n* = 6; Japan, *n* = 2; Korea, *n* = 1; and Hong Kong, *n* = 1). In addition, two studies each came from Germany and Sweden, and one study was from Macedonia. For the analysis, the authors created a combined estimate outcome variable, which included the self-reported outcomes “falling asleep”, “awakenings” and “sleep disturbances”. For the association between noise levels at night and the risk of HSD (combined estimate), L_night_ was included as a continuous variable from 40–65 dB. The noise-related risk increase per 10 dB L_night_ was OR = 2.13 (95% CI 1.82–2.48) for road traffic noise, OR = 3.06 (95% CI 2.38–3.93) for rail traffic noise and OR = 1.94 (95% CI 1.61–2.33) for aircraft noise. On the basis of this systematic review, the WHO published recommendations for traffic noise exposure levels. The relevant increase of the absolute risk considered for setting the recommended noise limits was 3% for HSD [13]. Based on the predicted 3% prevalence, the recommended nighttime traffic levels are below 45 dB L_night_ for road traffic, below 44 dB L_night_ for rail traffic and below 40 dB L_night_ for aircraft noise. Based on the WHO guidelines, Dhzambov and colleagues [15] estimated the burden of disease for Bulgaria to be 15,468 DALYs for severe sleep disturbance from road, rail traffic and aircraft noise.

Exposure-response functions for noise-related HSD based on European studies are scarce. Recently, the Swiss SiRENE study showed exposure-response functions for HSD from road traffic, rail traffic and aircraft noise [16]. The strongest association was found for aircraft noise followed by rail traffic and road traffic noise. In addition, Lechner and colleagues [17] established exposure-response functions for aircraft, rail traffic and road traffic noise (with subgroup motorway noise) (L_night_) for the city of Innsbruck, Austria. Again, the strongest associations were found for aircraft noise and the weakest for rail traffic noise. A study on the Atlanta airport showed adverse effects of aircraft noise on sleep, i.e., a significant exposure-response function for awakening probability and maximum sound pressure level [18,19]. Statistically significant exposure-response curves for short-term annoyance and nocturnal aircraft noise have also been shown using data from the NORAH study [20]. In addition, higher insomnia and daytime hyperinsomnia have been observed in residents exposed to higher aircraft levels [21]. A study on three French airports showed that aircraft noise was associated with a shorter total sleep time and more feelings of tiredness while awakening [22].

The aim of this study was to describe the relationship between HSD and transportation noise for Leipzig, a major city in Germany, and to establish exposure-response functions. We used address-specific road traffic data based on the primary and secondary road networks, rail traffic (including railway and tram) and aircraft traffic data. This survey was part of a study on the effects of transportation noise on mental disorders [23].

## 2. Materials and Methods

### 2.1. Study Design and Population

We used the address and questionnaire data from the LIFE-Adult cohort study. The study started in 2011 and is conducted by the Leipzig Research Centre for Civilization Diseases (LIFE). The LIFE study is done in accordance with the Declaration of Helsinki. Each participant provided written informed consent.

The study population includes 10,000 randomly selected residents (mainly ≥ 40–79 years old, and an additional subset of 400 individuals aged 18–39 years) living in Leipzig, Germany. The response was 31% for the baseline assessment. More information on the study objective, design and procedures is given in [24,25].

For the present analysis, we used questionnaire data from the second wave, which was conducted from June 2018 to December 2021. A total of 6115 persons participated in Wave 2, resulting in a loss-to-follow-up value of 39%. Between baseline and follow-up, some participants moved to the surrounding countryside of Leipzig. For the analysis, we only included participants living in the city of Leipzig during Wave, leading to a total of *n* = 5670 participants.

### 2.2. Outcome Assessment

For the second wave, an additional questionnaire on sleep disturbances from road, rail and aircraft traffic noise was included. The questionnaire was developed as part of the NORAH study on disease risks [26]. Self-reported sleep disturbances from individual traffic noise sources when falling asleep, during sleep and while waking-up during the last 12 months were determined according to the internationally standardized ISO/TS15666:2003 [27]. Answers were given on a five-point rating scale ranging from “not” (1) to “extremely disturbed” (5). For the analysis, a mean score over all three questions was calculated [28]. Participants characterized by a score of ≥4 (“very”and “extremely” disturbed) were defined as being “highly sleep disturbed” (HSD) [29]. The questionnaire is widely used in research on noise-related sleep disturbances. Results based on the questionaire form the basis of the current environmental guidelines for the European region of the WHO [14,30].

### 2.3. Noise Exposure Assessment

Residential exposure to road, rail and air traffic was determined for the most exposed façade at 4 m height for the reference year 2012 using participants’ individual addresses provided in the baseline assessment. The official noise map of the city of Leipzig is provided in Appendix A.

#### 2.3.1. Road Traffic Noise

Traffic input data for modeling road traffic noise on the main road network were based on calculations conducted by the Environmental Protection Agency in Leipzig according to the European Environmental Noise Directive (END). Thus, georeferenced annual data on the average daily traffic volume (AADT) and the proportions of heavy vehicles (>3.5 tongross vehicle weight) were available for all segments of the main road network. These AADT values were differentiated by day, evening and night hours according to the then-current German noise calculation guideline “Vorläufige Berechnungsvorschrift für den Umgebungslärm an Straßen” (VBUS) [31]. Other features of the main road network that are relevant for noise modeling (e.g., road surfaces, maximum permissible speeds) were also taken from the model used for strategic noise mapping.

Noise mapping according to the END is generally restricted to the main road network. However traffic noise on the secondary road network also contributes to the overall noise load in urban environments [32]. Becker [32] showed, for an exemplary urban residental quarter in the major city Dresden, Germany, that traffic noise levels increased by sometimes more than 15 dB when including traffic volumes from the secondary road network in the noise modeling. Thus in real life, traffic noise exposure is likely to be higher than noise levels calculated for the strategic noise maps according to the EU Environmental Noise Directive (2002/49/EC).

Data on traffic volume in the secondary road network were not available for Leipzig, and therefore had to be estimated. For this, the total mileage of all vehicles on the road network in Leipzig in 2012 (in vehicle kilometers) was estimated based on information on the size and composition of the local vehicle fleet in 2012, according to the statistics of the German Federal Motor Transport Authority [33,34] and statistics on the German average annual mileage per vehicle class in 2013 [35]. Traffic volume on the secondary road network was then calculated as the difference between the estimated total vehicle mileage in Leipzig and the sum of the mileage on the main road network taken from the END noise model input data. The average daily traffic volume for the secondary road network was estimated to be around 1600 cars per day. According to Becker [32], who used traffic loads with manual counts, there is a large range in daily traffic values within the secondary road network, especially between collector roads and local streets. Therefore, traffic volume in the secondary road network of Leipzig was differentiated between collector roads and local streets based on the ratio of the counted traffic volume in the analysis by Becker [32]. For Leipzig, we estimated the average daily traffic volume to be around 780 vehicles per day for local streets and around 2600 vehicles per day for collector roads.

Since there was no data available on the traffic volume of heavy vehicles in the secondary road network of Leipzig, the proportions recommened for municipal roads in the German calculation method VBUS [31] were used as an approximation. Additionally, for all secondary roads, a maximum legal speed limit of 50 km/h and a road surface of non-corrugated mastic asphalt, asphalt concrete or grit-mastic asphalt has been assumed. Road traffic noise source emissions and sound propagation were modeled according to VBUS [31] using the software SoundPLANnoise, version 8.1 (SoundPLAN GmbH, Backnang, Germany).

#### 2.3.2. Railway and Tram Noise

##### Railway Traffic Noise

Detailed information on railway vehicle movements was provided for a representative weekday in 2012 (19 April 2012) by the DB Netz AG (the main German railway infrastructure manager) for all trains running in and through the city area of Leipzig on that day, including light and commuter rail, regional, inter-city and high-speed rail as well as freight trains and other train movements. Train movement data was georeferenced and attributed to individual track sections of the rail network GIS model of the city of Leipzig with the software ESRI ArcGIS Version 10.7 and 10.8 (ESRI Deutschland GmbH, Kranzberg, Germany). Railway noise was modeled according to the German calculation method Schall 03 [36].

##### Tram Traffic Noise

Data on noise emission levels at individual rail track segments was provided directly by the Environmental office of the city of Leipzig. This data was modeled according to the German calculation method VBUSch [37] for the 2012 strategic noise map of Leipzig.

Noise source emissions and sound propagation were modeled with the software SoundPLANnoise. For the main analysis, train and tram noise expositions were combined through energetic addition to represent railway and tram noise (labeled as “rail traffic” noise). In an additional analysis, we estimated the isolated effect of railway noise on HSD.

#### 2.3.3. Aircraft Noise

Input data for aircraft traffic noise modeling was taken from the data acquisition system VBUF-DES [38] of the Leipzig/Halle airport for the year 2011. It was provided by the Saxon State Office for the Environment, Agriculture and Geology and corresponds to the input data of the 2012 END strategic noise map. Aircraft noise immission levels were calculated according the the German calculation method VBUF-AzB [39] with the software program CadnaA (DataKustik GmbH, Gilching, Germany).

#### 2.3.4. Calculation of Acoustical Parameters

To determine the noise exposure of study participants, façade noise levels were calculated at the physical adress of the participants. Receptor points were modeled at a uniform height of 4 m above the ground at the residential buildings under investigation. The exposure assessment was then based on the maximum value of the calculated noise levels at the receptor points of each individual building.

Long-term average sound levels L_Day,06–18h_, L_Evening,18–22h_, and L_Night,22–06h_ were determined for all three traffic types (road, rail and aircraft). These three noise indicators represent A-weighted energy-equivalent mean continuous sound levels over the respective averaging period. The noise index L_den_, representing a weighted day–evening–nightnoise index over the 24-h day (with an additional 10 dB increase for night hours and a 5 dB increase for evening hours) was derived from the beforementioned noise indicators. We used the L_den_ (day–evening–night noise level) and the L_night_ (average sound level between 10 p.m.–6:00 a.m.) for studying the effect of transportation noise on sleep disturbance.

In contrast to the more continous road traffic noise, rail and aircraft traffic causes rather intermittent noise situations with dominant single noise events and relatively calm time periods in between. These “noise peaks” may also have an influence on the perception of noise exposure and possibly also have damaging health consequences [40]. Consequently, for all nightly aircraft overflights and train movements, the maximum A-weighted sound level L_max_ at the receptor points at the building façade were calculated. The Number Above Threshold (NAT) level frequency criterion was then used as a noise indicator to express the impact of the individual noise events. In the NAT criterion, the number of noise events N is specified which exceed a certain threshold value Lp,threshold (NAT (Lp,threshold) = N). In the Noise Related Cognition and Health (NORAH) Study on disease risks [23], the threshold value that is exceeded N times was chosen in accordance with the criteria for establishing night protection zones around airports as determined in the German Act for Protection against Aircraft Noise [41]. This procedure was also followed in the present study. The calculated NAT-6 value determines the immission level at the most exposed receptor point that is exceed 6 times. In this study, NAT-6 levels for railway (tram excluded) and aircraft noise in the time period from 10 p.m. to 6 a.m. were considered.

### 2.4. Confounders

All models were adjusted for age, sex and socioeconomic status (SES). Age was included as a quadratic term and sex as a two-category variable (female or male) in the analyses. For the SES, an index was determined according to Lampert et al. 2013 [42], which is based on three dimensions: education (school and vocational qualifications), occupational status, and income (net equivalent income). For our analyses, the SES index is grouped into three categories: low (≤9.2 points), medium (˃9.2–15.3 points), and high (˃15.3 points).

### 2.5. Statistical Analysis

We used pseudonymized data for statistical analysis. Multivariate logistic regression analysis was performed to calculate odds ratios (ORs) and 95% confidence intervals for each traffic noise source (road, rail and aircraft traffic). The effect of transportation noise was analyzed by grouping each traffic noise source in 5 dB categories. The noise level below 40 dB was defined as the reference category. In addition, participants with continuous aircraft noise levels below 40 dB but with at least 6 maximum nightly levels above 50 dB (NAT-6) formed a separate exposure category. Analyses for all traffic noise sources were performed with continuous values of sound levels (per 10 dB). In further analyses, quadratic functions for the percentage of highly sleep disturbed (%HSD) were created for all three traffic noise sources (road, rail and aircraft) for L_den_ and L_night_ noise levels starting at ≥35 dB. The increase in HSD per 1 dB traffic noise was determined by the prevalence ratios which represent the proportion of highly sleep disturbed individuals divided by the proportion of those not highly sleep disturbed at a given exposure level. The calculated exposure-response-relationships were compared with the combined estimate results of the review by Basner and McGuire [14]:

For this, we used the following equations for combined estimates:Road %HSD = 19.4312 − 0.9336 × L_night_ + 0.0126 × (L_night_)^2^
Rail %HSD = 67.5406 − 3.1852 × L_night_ + 0.0391 × (L_night_)^2^
Aircraft %HSD = 16.7885 − 0.9293 × L_night_ + 0.0198 × (L_night_)^2^

Basner and McGuire [14] determined the exposure-response curves using the results from the logistic regressions of the individual studies. Thus, they used odds rather than proportions (or prevalence). ORs overestimate relative risk ratios (RRs) a prevalence of over 10%. Therefore, we expected the calculated odds would be higher compared with prevalence at higher noise levels. In a sensitivity analysis, we compared the exposure-response relationship from our logistic regression with that of the WHO using the same methodology. For this, univariate logistic regressions were modeled using only L_night_ as the independent variable. Then, the resulting predicted values were modeled in 1 dB steps with second-order polynomials.

All analyses were performed with Stata Version 17.0 (StataCorp LLC, College Station, TX, USA).

## 3. Results

### 3.1. Descriptive Characterization of Study Population with Regard to Traffic Source

Approximately 2.7% of participants were HSD from road traffic noise, 1.2% from rail traffic noise and 2.0% from aircraft noise. Slightly more women tended to be HSD from road and aircraft noise than men (road: 3.1% versus 2.2%; air: 2.3% versus 1.7%). For rail traffic noise, the proportion of HSD was 1.1% in females and 1.7% in males. With regard to age, participants aged between 50–59 years had the highest proportion of aircraft noise-related HSD (about 3.0%). For road and railway traffic, the highest HSD proportion was observed for the age class 40–44 years (road: 5% and rail: 3.8%), though the sample size was very low. Moreover, participants with a low SES were characterized by a slightly higher proportion of road- and rail traffic-related HSD than participants with a moderate and high SES (road—low SES: 3.7%, moderate SES: 2.7% and high SES: 1.9%; rail—low SES: 2.0%, moderate SES: 1.5% and high SES: 0.9%). There was no difference for aircraft noise-related HSD with respect to SES classes. Participant characteristics are summarized in Appendix A.

### 3.2. Risk for HSD and Traffic Noise Exposure

We observed an increase in HSD risk with increasing road traffic and rail traffic (Table 1). For HSD from road traffic noise, the risk estimates, assuming a linear function, were OR = 2.81 (95% CI 1.88–4.21) per 10 dB for the L_den_ and OR = 2.86 (95% CI 1.92–4.28) per 10 dB for the L_night_. Comparably high risks were also found for rail traffic noise (L_den_: OR = 2.68, 95% CI 1.92–4.28 per 10 dB; L_night:_ OR = 2.67, 95% CI 2.03–3.50 per 10 dB). The strongest association between the risk of HSD and traffic noise was found for aircraft noise (L_den_: OR = 13.06, 95% CI 9.25–18.44 per 10 dB; L_night_: OR = 19.66, 95% CI 11.47–33.71 per 10 dB; Table 1). On the basis of categorical analyses, a HSD risk increase with increasing aircraft noise was observed up to 55 dB L_den_ and up to 50 dB L_night_. Thereafter, HSD risk dropped slightly but was still very high, though sample sizes were very low at higher noise level categories.

### 3.3. Comparison with the WHO Curves

For comparison of exposure-response- relationships between our study and the values reported in the WHO review, we derived the following formulas for the LIFE-Adult study, starting at 35 dB:Road %HSD = 20.5376 − 1.0010 × L_night_ + 0.0129 × L_night_^2^
Rail %HSD = 17.8978 − 0.9177 × L_night_ + 0.0122 × L_night_^2^
Air %HSD = −321.4811 + 14.0045 × L_night_ − 0.1365 × L_night_^2^

Figure 1 shows the exposure-response-curves for the %HSD for all three noise sources from both the LIFE-Adult study and the WHO review by Basner and McGuire [14]. For road traffic noise, curves were not directly comparable because we additionally included the secondary road network. For rail traffic noise (i.e., railway and tram traffic noise), the calculated proportion of people with HSD was lower in the LIFE-Adult study compared with the calculated curve of the WHO review [14]. A similar curve was obtained for the LIFE-Adult study when considering the exposure-response- association between HSD and railway noise only (without tram noise) (Appendix A): The HSD prevalence was 1% at 35 dB, 1% at 45 dB, 4% at 55 dB and 9% at 65 dB (Appendix A). The 3% HSD threshold from railway noise was reached at 53.1 dB L_night_.

For road and rail traffic noise, the calculated 3% HSD threshold (set in the WHO EU guidelines) was reached at 45 dB L_night_ for the WHO and at 51 dB L_night_ for the LIFE-Adult study.

For aircraft noise, the proportion of HSD was considerably higher in the LIFE-Adult study than according to the WHO review. The %HSD increased from 1% at 35 dB to 32% at 45 dB in the LIFE-Adult study. There are differences in the proportion of HSD for 40 dB L_night_ between the modeled %HSD and the observed %HSD for the LIFE-Adult study: the observed proportion of HSD was 1% for aircraft noise levels of <40 dB with maximum (NAT-6) levels of <50 dB and 9% for the category < 40 dB with maximum (NAT-6) levels of 50 dB and more (see Table 1). The modeled proportion of HSD was 20% at 40 dB L_night_ aircraft noise.

For the WHO curves, models were based on noise levels between 40 and 60 dB L_night_. According to the WHO formula, 15% are HSD at 45 dB aircraft noise. Results of the modeled %HSD for the LIFE-Adult study and the WHO review for the traffic noise levels 35 dB, 45 dB, 55 dB and 65 dB are shown in Table 2.

In addition, when using the “WHO method” (i.e., odds of HSD obtained by the predicted logistic regression values, the odds of HSD were similar to those obtained directly from the study data points for road traffic noise. Thus, %HSD were lower in the LIFE-Adult study (odds and prevalence) compared with the WHO review. For rail traffic and aircraft noise, we observed higher odds of being HSD at higher noise levels (i.e., for rail traffic: 65 dB and for aircraft traffic: 55 dB) when using the predicted logistic regression values compared with the prevalence values obtained directly from the study data points (rail %HSD: 15% versus 10%; air %HSD: 47% versus 37%). Results are shown in Appendix A.

## 4. Discussion

In the present study, we investigated the risk of being HSD from different traffic noise sources in the population-based LIFE-Adult study. About 2.7% of participants were classified as being HSD from road traffic noise, 1.2% from rail traffic noise and 2.0% from aircraft noise. Similar prevalence has been reported by Lechner and colleagues for the city of Innsbruck [17], where the %HSD was 5.5% for road traffic, 1.5% for rail traffic noise and 3.9% for aircraft noise. In the Swiss SIRENE study, a higher HSD prevalence was observed for all traffic sources: the proportion of HSD was 9.3% for road traffic noise, 10.0% for rail traffic noise and 12% for aircraft noise. In addition, for Barcelona, the road traffic-related %HSD was 9.9% [43]. Two Asian studies reported an HSD prevalence of 4.1% [44] and 5.1% [45] for road traffic noise.

In our study, the risk of HSD was statistically significantly associated with all traffic noise sources for the L_night_ and the L_den_. The strongest HSD risk was found for L_night_, followed by L_den_. Janssen and colleagues [46] stated that L_night_ adequately reflects the number of aircraft noise events. Nevertheless, it might be important to consider the number of events in addition to L_night_, especially at higher maximum noise levels [46,47].

The highest risk for traffic noise-related HSD was found for aircraft noise: the risk increase was OR = 19.66, 95% CI 11.47–33.71 per 10 dB increase in L_night_. For road and rail traffic, similar risk estimates were observed (road: OR = 2.86, 95% CI 1.92–4.28; rail: OR = 2.67, 95% CI 2.03–3.50 per 10 dB L_night_ increase). Our results are in line with findings from other European cities, which also observed a stronger association for aircraft noise on the HSD risk compared with road and rail traffic [16,17,48]. This indicates that aircraft noise is particularly sleep disturbing. Using data of over 50 original studies, Miedema and Oudshoorn [49] showed that the proportion of highly annoyed participants was higher for aircraft noise in comparison with road and rail traffic noise at similar noise levels. As a consequence, aircraft noise might be particularly harmful to mental health. A recent systematic review showed that the depression risk was increased by 12% per 10 dB aircraft noise but “only” by 2–3% for road traffic and rail traffic noise [50].

Our exposure-response functions differed from the published WHO curves. For road traffic noise, we were not able to directly compare our results with the WHO curves since we also included the secondary road network. A 3% HSD prevalence is found at 45 dB L_night_ according to the WHO and at 51 dB L_night_ according to the LIFE-Adult study. But this does not mean that road traffic noise is less sleep disturbing in the LIFE-Adult study than according to the WHO review. If only the primary road network is taken into account, the prevalence of HSD is higher at a given noise level. Moreover, the level of road traffic noise exposure might be underestimated when only considering the primary network as an exposure source. Indeed, Becker [31] showed that noise levels are up to 15 dB higher if the secondary road network is taken into account compared with considering primary road networks only. In addition, by considering side roads and measurements outside of conurbations, the number of people affected increases. It has been shown that the number of people classified as exposed to L_den_ > 55 dB increased by up to 119% outside of conurbations and 5% inside conurbations [51]. We encourage further research to incorporate the side road network when investigating health effects related to road traffic noise.

For rail noise, the 3% threshold was reached at 45 dB in the WHO review and at 51 dB in the LIFE-Adult study. To the best of our knowledge, the WHO exposure-response curve is based on four studies investigating noise from train traffic in cities and around railway tracks from one city to another. In our study, we considered rail and tram traffic noise in an urban area in our main analysis. When the association between HSD and only railway noise was also studied, a similar exposure-response curve was observed for the LIFE-Adult study. The 3%HSD prevalence was reached at 53.1 dB when considering railway noise only. Our results are comparable to the results of the Innsbruck study, where the threshold was reached at 49 dB [17]. In the Swiss SiRENE study, 2% were HSD at 35–40 dB and 7% at 40–45 dB [16]. Differences in HSD prevalences may arise from different study settings. Moreover, different definitions of L_night_ were used. We defined L_night_ as the time from 10 pm to 6 pm; the SiRENE study used the time window from 11 pm to 7 am. Future studies investigating the effect of different L_night_ definitions on sleep disturbance prevalences are desirable.

For aircraft noise, our exposure-response function was considerably higher compared with WHO (LIFE: 45 dB, 32% and 55 dB, 36%; WHO: 45 dB, 15% and 55 dB, 26%). Our results are comparable to the results of the Innsbruck study [17] and the Swiss study [16] for 55 dB L_night_. In both studies, about 40% of participants were HSD at this noise level. At 45 dB aircraft noise, the proportion of HSD was about 20%, and thus lower compared to the LIFE-Adultstudy.

There might be several reasons for the difference. The number of annual flight movements at Leipzig/Halle airport increased from approximately 60.000 in 2010 to approximately 80.000 in 2019. In the year 2019, there were on average 216 take-offs and landings per day at Leipzig/Halle airport. About half of the flight movements occurred between 10 p.m. and 6 a.m. and consisted mainly of airfreight traffic (details are provided in the supplement of the report [23]). Thus, Leipzig/Halle airport may be considered a “high-rate-change- airport”. These airports are characterized by higher annoyance at given noise levels than “low-rate-change airports” [52].

In addition, large military transport airplanes of the Antonov type (AN 12 and AN 26) built in the 1950s–1960s, are also used by the German Armed Forces, automotive suppliers and others for freight transport [53]. Thus, noise from military planes might be perceived as more disturbing, especially at lower noise levels. Gelderblom and colleagues [54] found a higher proportion of highly annoyed residents at lower L_dn_ levels around an airport with mixed civil and military flight operations compared with a civil airport in Norway. They observed that fighter jets were responsible for relatively high noise levels per event in areas characterized by a low L_dn_. This may also apply to our study, where most participants were living in areas characterized by low aircraft noise levels. Moreover, the higher proportion of aircraft noise-induced HSD in the LIFE-Adult study compared to the WHO review may be a result of increased HSD over time; similar observations have been made for annoyance [55]. The WHO review mainly includes older studies that may underestimate the current proportion of people with HSD at a given noise level. Recently, the WHO review was updated by Smith and colleagues [30] who included 13 new studies on noise-related HSD. They also updated the exposure-response functions, which were comparable to the 2018 review for road and rail traffic noise. For aircraft noise, the exposure-response curve was characterized by a higher proportion of HSD. The proportion of HSD was about 18% at 45 dB and 30% at 55 dB. Altogether, the exposure-response curves for aircraft noise are still lower compared to our study, and the Innsbruck and SIRENE study (which was included in the update). The update also included studies characterized by a very low response of 10% and lower (e.g., [19,48]), which may have biased the results. A re-analysis of exposure-response function with regard to study quality is desirable.

In addition, the WHO recommends noise levels below 40 dB for nighttime aircraft noise [13]. In our study, we observed 2% HSD at 35 dB which increased up to 20% HSD at 40 dB L_night_. Our results are in line with the Austrian [17] and Swiss study [16] which also observed elevated %HSD at low aircraft nightly noise levels. The results suggest that aircraft noise per se is perceived as particularly disturbing at very low levels [40]. Based on the results of our study (as well as the SiRENE and Innsbruck studies), we recommend reducing nightly aircraft noise exposure levels to 35 dB L_night_.

### Strengths and Limitations

The LIFE-Adult study was population-based, however, response at the study baseline was rather low (33%) and loss-to-follow-up was rather high; only 61% of Wave 1 participants participated in Wave 2. According to a recent non-responder analysis [25,56], compared with the general population of the city of Leipzig and non-participants, LIFE-Adult participants were older, slightly more often male, had a higher socioeconomic status, a healthier lifestyle and were healthier. However, we found no indication of substantial age-, sex-, or social status-related differences in the risk of reporting being highly sleep disturbed. Moreover, all analyses were adjusted for age, sex and socioeconomic status. We therefore consider a considerable age-, sex-, or social state-induced selection bias to be rather unlikely.

In addition, data of Wave 2 was collected before and during the SARS-CoV2 pandemic. The study was interrupted for some time during the pandemic, and data collection was hampered. This resulted in a pre-, between and post-pandemic data collection. In addition, the SARS-CoV2 pandemic may have influenced the self-rated mental health and sleep outcome ratings. It has been shown that mental health symptoms were increased soon after the pandemic outbreak [57]. However, by mid-2020 the symptoms decreased to pre-pandemic levels [57]. In addition, the effect of the SARS-CoV2 pandemic was also studied in the LIFE study for depression and anxiety, and no statistically significant effect was observed [23]. Therefore, we regard a strong effect of the SARS-CoV2 pandemic on our results as rather improbable.

%HSD values were determined from June 2018 to December 2021, and exposure data originate from 2012. In principle, to avoid cause and effect bias, it is required that the exposure precedes the outcome. However, sleep disturbances should be more related to the current noise situation than to long-term “chronic” exposure conditions. In fact, we used the most recent noise maps available at the time of the study. In 2012, strategic noise maps were prepared by the city of Leipzig according to the EU Environmental Noise Directive (2002/49/EC). In the year 2017, new strategic noise mappings were performed but results and the noise action plan were only published by mid 2022. Within the study context, it was not possible to assess the impact resulting from the time delay between noise exposure modeling and data collection during the follow-up for each individual participant. However, with the available data, we conducted an analysis for road and air traffic development in Leipzig between 2012 and the time of the follow-up-survey in order to qualitatively describe the uncertainty in the noise exposure values. The results show that road traffic noise was relatively constant over time [23]. For aircraft noise, Leipzig experienced an increase in the number of flight movements and a number of aircraft noise reduction measures, which led to a slight decrease in noise levels [58]. Thus, aircraft noise exposure might be slightly overestimated. Further details on the assessment of noise exposure uncertainties are given in the supplementary of the final project report [23].

Moreover, we did not include information on residential floors in our noise models, which may lead to bias in exposure estimation. However, current research suggests that residential floor may exert only little influence [59]. Vienneau and colleagues showed that noise levels generally do not substantially decrease with floor height [59]. In addition, we did not have information on indoor noise levels, and did not consider bedroom orientation and window position in our analysis, which may lead to uncertainties in noise exposure assessment [60].

Further, we did not assess the effect of combined noise exposure from different sources on HSD risks. This is because we wanted to compare our dose-response curves with the WHO curves of the environmental noise guidelines which currently also do not consider combined noise effects. Indeed, several models for the health effects of combined traffic noise have been proposed [61,62,63,64], and a recent study suggests that conventional energetic summation of noise pressure levels might underestimate the noise effects on health [64].

As a further limitation, there were only a small number of subjects exposed to higher aircraft noise levels. This may have led to uncertainties in the determined exposure-response curves, which ultimately represent extrapolations for high level values. Moreover, in this study we determined self-reported sleep disturbances from individual traffic noise sources by mentioning the traffic sources in the questions. The WHO-review by Basner and McGuire [14] showed that the association between traffic-related sleep disturbances were much weaker when the question did not refer to a specific noise source. However, this observation is based on a limited number of studies (aircraft, *n* = 1; road, *n* = 3; rail, *n* = 2). Moreover, we did not consider orientation of bedrooms in our analysis as a possible effect modifier.

In addition, our questionnaire used is internationally standardized but does not present a diagnostic tool for sleep disorders. In the LIFE-Adult study, (non-traffic-related) sleep disorders were determined by the Pittsburgh Sleep Quality Index [65]. About 2.1% reported having a very bad subjective sleep quality and about 3.3% of participants reported a sleep duration of less than 5 h. In addition, prevalence of sleep disorders might be higher in larger cities than in rural areas. However, according to the German DEGS1 study, similar levels of sleep disorder prevalence of about 6% have been observed for rural areas (with less than 5000 inhabitants) and large cities with more than 100,000 inhabitants [66].

The strength of our study lies in the detailed assessment of traffic noise exposure by including the secondary road network and tram noise in our noise models. In addition, exposure was determined at low noise levels starting from 35 dB. Moreover, we used a standardized questionnaire and cut-off values for the definition of HSD [27,29]. Our analyses were adjusted for age, sex and socio-economic status to account for potential confounding.

## 5. Conclusions

In conclusion, the Leipzig LIFE-Adult study revealed a strong association between traffic noise and sleep disturbances. Harmful effects on sleep were found in particular for aircraft noise followed by road and rail traffic noise. We observed a considerable proportion of highly sleep disturbed persons at low aircraft noise levels. Based on our results, we suggest reconsidering threshold values for nightly aircraft exposure. In addition, as a methodological improvement over previous studies, we included the secondary road network and tram into traffic noise modeling. Future studies on the association between road traffic noise and sleep disturbances should also apply these methodological improvements to derive the exposure-risk relationship more precisely.

## Figures and Tables

**Figure 1 ijerph-20-04903-f001:**
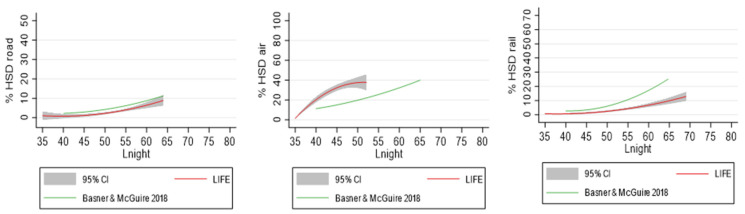
Comparison of HSD risk curves for road traffic, rail traffic and aircraft noise between the LIFE study and the WHO (Basner and McGuire 2018 [14]).

**Table 1 ijerph-20-04903-t001:** Multivariate logistic regression for the risk of HSD from road traffic, rail traffic and aircraft noise.

Noise Measure	Road Traffic Noise				Rail Traffic (Railway and Tram Combined)				Aircraft Noise			
	Total	HSD *n* %	OR	95%CI	Total	HSD *n* %	OR	95%CI	Total	HSD *n* %	OR	95%CI
**L_den_**												
<40 dB					576	3 0.5%	1	Reference				
<40 dB, max. <50 dB									4421	34 0.8%	1	Reference
<40 dB, max. ≥50 dB ^1^									102	3 2.9%	3.82	1.15–12.7
≥40–<45 dB					1075	3 0.3%	0.53	0.11–2.66	347	31 8.9%	12.5	7.56–20.7
≥45–<50 dB					1232	13 1.1%	2.00	0.57–7.04	93	20 21.5%	35.3	19.4–64.4
<50 dB	176	2 1.1%	1	Reference	1091	16 1.5%	2.80	0.81–9.65				
≥50–<55 dB	369	4 1.1%	0.95	0.17–5.27					21	9 42.9%	91.1	35.6–233.2
≥55–<60 dB	1438	28 1.9%	1.48	0.25–6.34	499	8 1.6%	3.10	0.82–11.8	11	4 36.4%	69.7	19.3–251.1
≥60 dB					471	27 5.7%	11.4	3.42–37.9				
≥60–<65 dB	1988	64 3.2%	2.41	0.58–9.98								
≥65–<70 dB	1048	35 3.3%	4.24	1.02–17.6								
≥70 dB	85	4 4.7%	4.03	0.72–22.6								
continuously (per 10 dB)			2.81	1.88–4.21			2.68	1.92–4.28			13.06	9.25–18.44
**L_night_**												
<40 dB	171	2 1.2%	1	Reference	2343	14 0.6%	1	Reference				
<40 dB, max. <50 dB									4536	41 0.9%	1	Reference
<40 dB, max. ≥50 dB ^1^									361	32 8.9%	10.5	6.51–16.9
≥40–<45 dB	407	4 1.0%	0.83	0.15–4.60	1109	16 1.4%	2.30	1.16–4.93	70	16 22.9%	32.1	17.0–60.8
≥45–<50 dB	1704	28 1.6%	1.42	0.33–6.02	844	9 1.1%	1.77	0.76–4.12	17	8 47.1%	91.6	33.3–251.6
≥50–<55 dB	2014	64 3.2%	2.75	0.67–11.4	306	8 2.6%	4.45	1.85–10.7	11	4 36.4%	59.4	16.6–212.7
≥55–<60 dB	737	35 4.7%	4.11	0.98–17.3	245	14 5.7%	9.66	4.54–20.6				
≥60 dB	71	4 5.6%	4.68	0.83–26.3	97	9 9.3%	17.3	7.25–41.5				
continuously (per 10 dB)			2.86	1.92–4.28			2.67	2.03–3.50			19.66	11.47–33.7

^1^ is equivalent to NAT-6 Model adjusted for age (quadratic term), sex (male/female) and socioeconomic status (3 categories).

**Table 2 ijerph-20-04903-t002:** Example for %HSD from road traffic, rail traffic and aircraft noise for the LIFE-Adult study and the WHO Environmental Noise Guidelines for the European Region: A Systematic Review on Environmental Noise and Effects on sleep [14].

L_night_	%HSD from Road Traffic Noise		%HSD from Rail Traffic Noise		%HSD from Aircraft Noise	
	WHO	LIFE-Adult	WHO	LIFE-Adult	WHO	LIFE-Adult
35 dB	- ^1^	1	- ^1^	1		1
45 dB	3	2	3	1	15	32
55 dB	6	5	11	4	26	36
65 dB	12	10	26	10	40	- ^2^

^1^ The WHO models had a noise level limit of 40 dB. ^2^ None of the participants in the LIFE-Adult study was exposed to aircraft noise levels of 52 dB and more.

## Data Availability

Restrictions apply to the availability of these data. Data was obtained from the Leipzig Research Center for Civilization Diseases (LIFE) and are available on request from the LIFE research center (https://www.uniklinikum-leipzig.de/einrichtungen/life accessed on 1 March 2023).

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
