# Peer review of "Traffic-Related High Sleep Disturbance in the LIFE-Adult Cohort Study: A Comparison to the WHO Exposure-Response-Curves"

_ijerph, 2023, doi:10.3390/ijerph20064903_

Round 1

Reviewer 1 Report

The article is of interest due to the excellent review of the representative parameters of noscturnal noise and the exhaustive record of the values of said parameters.

The questionnaire used to  assess the study outcome (an adaptation of the NORAH questionnaire) at first sight does not seem very solid. It would have to be reinforced with a classic sleep quality questionnaire such as the PSQI.

Another piece of information that would give solidity to the article will be providing some type of information on the prevalence of sleep disorders (with special emphasis in insomnia symptoms) diagnosed in the health services of the studied area (Leipzig) in comparison with other nearby areas, for example of rural typology.

The impression is that there is an excellent work on environmental noise data collection but from my point of view it would not be strange if the reader did not find a consistent relationship between this and sleep disturbances.

Author Response

Dear reviewer, thank you very much for your kind words and suggestions. In research of environmental health risks, the questionnaire is widely used. The results of studies using this questionnaire form the basis for recommendations on noise limits by the WHO in the environmental noise guideline for the European Region. It is a validated and internationally standardized questionnaire.  We agree that the instrument is no diagnostic tool for the diagnosis of sleep disorders. Therefore, we have emphasized this in the limitations. Here, we have also included information on the prevalence of sleep disorders assessed with the PSQI and the difference in prevalences between rural areas and large cities.

“In addition, our questionnaire used is internationally standardized but does not present a diagnostic tool for sleep disorders. In the LIFE-Adult study, (non-traffic-related) sleep disorders were determined by the Pittsburgh Sleep Quality Index [66]. About 2.1% reported to have a very bad subjective sleep quality and about 3.3% of participants reported a sleep duration of less than 5h. In addition, prevalence of sleep disorders might be higher in larger cities than in rural areas. However, according to the German DEGS1 study, similar levels of sleep disorder prevalence of about 6% have been observed for rural areas (with less than 5000 inhabitants) and large cities with more than 100,000 inhabitants [67].”

Reviewer 2 Report

Dear authors, I liked your article, but I want to note that the first keywords are sleep and HSD. It seems to me that in the article it would be interesting to pay more attention to the impact of noise on the clinical aspects of concomitant sleep disorders. In the title, it was possible to change HSD to "subjective severity  of sleep disorders", since the term HSD reflects the content of an interesting article less accurately and is less understandable for clinicians and physiologists

Author Response

Dear reviewer, thank you very much for your kind words and suggestions. We have rephrased the key words to “self-reported high sleep disturbance”. Our questionnaire is validated and internationally standardized, but does not present a diagnostic tool for diagnosis of sleep disorders, we prefer to use “disturbance” instead of “disorder”. The abbreviation of “HSD” is widely used in environmental research on health effects of traffic, also by the WHO in their environmental noise guideline for the European Region.

Reviewer 3 Report

Overall:

This study aimed to describe the relationship between high sleep disturbances and transportation noise in Leipzig, Germany, and to establish exposure-response functions for aircraft, rail traffic and road traffic noise.

In general, the topic is interesting, but the manuscript lacks important information related to the methodology. This section should be improved and supplemented.

Introduction:

-          Introduction is mainly based on one reference – The WHO guidelines. It is necessary to expand the background of the analysed issue to justify the relevance of the study.

Materials and Methods:

-          Was the questionnaire for sleep disturbances developed based on approved and validated standards? Sleep disturbances are considered to be a symptom of mental illness, therefore it is important that questionnaire and methodology to assess sleep disorders is adapted by psychologists.

-          Was the sample of the study participants representative for the analysed city? Maybe the authors could provide comparison of the main characteristics of study participants with the whole population of Leipzig.

-          What could be the uncertainty and bias of the results taking into account that noise exposure assessment was performed in 2012, meanwhile the survey using questionnaire was conducted between 2018-2021? How could this impact the results and association between noise exposure and HSD? Information about railway vehicle movements was based on one day routes of 2012, meanwhile, the survey was conducted after a period of 6- 9 years, when there should have been significant changes in railway movements. How the authors account for these differences and changes? The same situation is with tram and aircraft data.

-          Was there a newer strategic noise map for Leipzig? 

-          Noise exposure for study participants was performed based on the most exposed façade at 4m height. Why was the floor of the participants’ residence not taken into account and the noise exposure assigned accordingly?

-          How has individual noise exposure from different sources and its effect on HSD been assessed when in many cases this exposure is cumulative or affects people living closer to the source? The methodology does not clearly state how exposure was determined and what percentage of participants lived closer to one or another noise source.

-          The authors mention that the GIS was used. It is necessary to specify more precisely what software and version was used in the study.

-          The map showing geographic location and area where the study was conducted would be useful.

Results:

-          Table should be provided representing demographic, socioeconomic and other characteristics of respondents.

Minor errors:

-          Modelling or modeling should be unified throughout the main text.

-          Line 169: 2.3.2. Railway railway and tram noise

Author Response

Dear reviewer, thank you very much for your kind words and your suggestions. Attached  you'll find the point by point response.

Round 2

Reviewer 3 Report

The authors considered comments and suggestions, most of which were included as limitations of the study. However, the manuscript has been slightly improved compared to the previous version.